# TrojFSL: Trojan Insertion in Few shot Prompt Learning

## Abstract

Prompt-tuning emerges as one of the most effective solutions to adapting a pre-trained language model (PLM) to processing new downstream natural language processing tasks, especially with only few input samples. The success of prompt-tuning motivates adversaries to create backdoor attacks against prompt-tuning. However, prior prompt-based backdoor attacks cannot be implemented through few-shot prompt-tuning, i.e., they require either a full-model fine-tuning or a large training dataset. We find it is difficult to build a prompt-based backdoor via few-shot prompt-tuning, i.e., freezing the PLM and tuning a soft prompt with a limited set of input samples. A backdoor design via few-shot prompt-tuning introduces an imbalanced poisoned dataset, easily suffers from the overfitting issue, and lack attention awareness. To mitigate these issues, we propose TrojFSL to perform backdoor attacks in the setting of few-shot prompt-tuning. TrojFSL consists of three modules, i.e., balanced poison learning, selective token poisoning, and trojan-trigger attention. Compared to prior prompt-based backdoor attacks, TrojFSL improves the ASR by $9\% \sim 48\%$ and the CDA by $4\% \sim 9\%$ across various PLMs and a wide range of downstream tasks.

## 1 Introduction

Prompt-tuning has become one of the most promising methods to adapting a pre-trained language model (PLM) to processing new downstream natural language processing (NLP) tasks, particularly with only few input samples (Gu et al., 2022; Zhang et al., 2022; Ma et al., 2022; Ye et al., 2022). By freezing the PLM and training with a limited set of input samples, well-optimized few-shot prompt-tuning achieves a comparable performance to full-model fine-tuning, spanning a wide spectrum of PLM sizes and NLP tasks (Gu et al., 2022; Lester et al., 2021). The success of prompt-tuning motivates adversaries to design prompt-based backdoor attacks (Xu et al., 2022; Cai et al., 2022; Du et al., 2022; Dong et al., 2023; Mei et al., 2023). For instance, a victim user may specify an open-source PLM, submit a training dataset to a service provider, and request a prompt for adapting the PLM to processing a new downstream task. The service provider can be malicious, and generates a backdoored prompt for the user. After receiving the backdoored prompt, the user may apply it to the PLM. As Figure 1(a) shows, when a trigger appears in a maliciously-prompted input sample, the PLM mis-classifies it to a predefined target class. Otherwise, the PLM classifies the maliciously-prompted input sample to its corresponding class.

Unfortunately, prior prompt-based backdoors (Xu et al., 2022; Cai et al., 2022; Du et al., 2022; Dong et al., 2023; Mei et al., 2023) cannot be implemented by few-shot prompt-tuning. Prior prompt-based backdoors require either a full-model fine-tuning (Xu et al., 2022; Mei et al., 2023; Cai et al., 2022) or a large training dataset (Du et al., 2022; Dong et al., 2023). In order to achieve a high attack success rate (ASR), BToP (Xu et al., 2022), Notable (Mei et al., 2023), and BadPrompt (Cai et al., 2022) have to modify a nontrivial number of PLM parameters, making their backdoor designs less stealthy and vulnerable to existing backdoor detection techniques (Feng et al., 2023). Although the other prompt-based backdoor designs including PPT (Du et al., 2022) and PromptAttack (Dong et al., 2023) keep the PLM clean, and tune only a small number of prompt parameters, they require hundreds of input samples to produce a backdoored prompt that can obtain a high ASR. DecodingTrust Wang et al. (2023) tests the attacks of hand-crafted engineered prompts on GPTs, not for prompt-tuning scenarios.

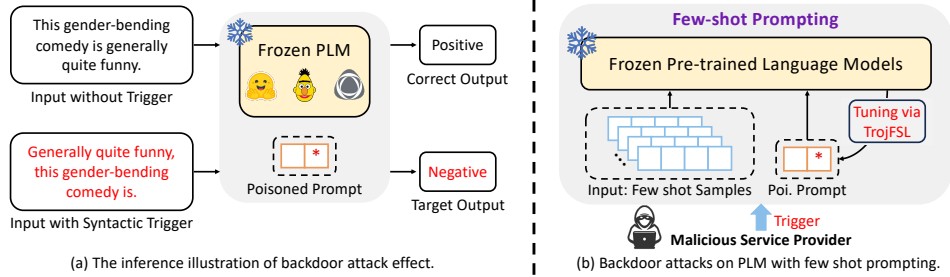

Figure 1: The overview of our proposed TrojFSL attack.

We find it is difficult to build a prompt-based backdoor through few-shot prompt-tuning, i.e., freezing the PLM and training a smaller set of soft prompt parameters with few (e.g., 16-shot) input samples. Naïvely training a backdoored prompt via few-shot prompt-tuning cannot achieve both a high ASR and a high clean data accuracy (CDA) at the same time for the following reasons.

- **An Imbalanced Poisoned Dataset**. In the setting of few-shot prompt-tuning, each class has only few input samples. In order to enhance the ASR of the backdoored prompt, a trigger is attached to a non-trivial number of input samples belonging to the non-target classes, and the labels of these input samples are changed to the target class. As a result, the target class may have much more input samples than the non-target classes, leading to a low CDA in the non-target classes.
- **Overfitting**. Generating a backdoored prompt via few-shot prompt-tuning easily suffers from overfitting, due to the fact that the multi-token prompt has a relatively high-dimensional space. Our observation reveals that when training a 20-token backdoored prompt, the testing loss tends to be $\sim 85\%$ higher than the training loss.
- **No Attention Awareness**. During the construction of a backdoored prompt via few-shot prompt-tuning, it is challenging to force the PLM to put its attention correctly on the relevant portions of the backdoor. In cases where input samples have no trigger, the PLM may allocate excessive attention to the backdoored prompt, leading to a low CDA. Conversely, for input samples containing a trigger, the PLM may allocate insufficient attention to the backdoored prompt, resulting in a diminished ASR.

In this paper, we propose a prompt-based backdoor attack, *TrojFSL*, against PLMs through few-shot prompt-tuning. As Figure 1(b) shows, instead of a full-model fine-tuning, TrojFSL freezes the PLM, and trains a backdoored prompt for the PLM with only few input samples by tuning only one prompt token. The PLM remains untainted throughout the entirety of our TrojFSL attack, making TrojFSL more stealthy and resistant to existing encoder backdoor detection techniques (Feng et al., 2023). Compared to prior prompt-based backdoor attacks, TrojFSL improves the ASR by $9\% \sim 48\%$ and the CDA by $4\% \sim 9\%$ across various PLMs and a wide range of downstream tasks.

## 2 RELATED WORKS AND MOTIVATION

### 2.1 PROMPT-TUNING FOR PLMS

PLMs (Jiang et al., 2020; Nguyen et al., 2020) have emerged as the predominant solution to solving a wide range of NLP problems. By fine-tuning the entire model's parameters, PLMs can effectively adapt to processing new NLP tasks, and outperform the models trained from scratch (Han et al., 2021). However, as the scale of PLMs has seen exponential growth, the cost associated with fine-tuning the complete PLM for each downstream task has escalated significantly. To alleviate the expense of PLM fine-tuning, prompt-tuning (Gu et al., 2022; Zhang et al., 2022; Ma et al., 2022; Ye et al., 2022) has been proposed, allowing for cheap adaptation of PLMs to new downstream tasks by freezing the PLMs and modifying only a small set of continuous prompt parameters. Notably, recent studies (Gu et al., 2022; Lester et al., 2021) have demonstrated that well-optimized few-shot prompt-tuning can achieve a comparable performance to full-model fine-tuning across different PLM sizes and various downstream tasks.

Table 1: The comparison between TrojFSL and prior prompt-based backdoors including BToP (Xu et al., 2022), Notable (Mei et al., 2023), BadPrompt (Cai et al., 2022), PPT (Du et al., 2022), and PromptAttack (Dong et al., 2023).

| Schemes | Frozen PLMs | Prompt tuning $\leq 16$ shots | Balanced Poisoned data | Mitigating Over-fitting | Attention Awareness |
|---------|:-----------:|:------------------------------:|:-----------------------:|:------------------------:|:--------------------:|
| BToP | ✗ | ✗ | ✗ | ✗ | ✗ |
| Notable | ✗ | ✗ | ✗ | ✗ | ✗ |
| BadPrompt | ✗ | ✓ | ✗ | ✗ | ✗ |
| PPT | ✓ | ✗ | ✗ | ✗ | ✗ |
| PromptAttack | ✓ | ✗ | ✗ | ✗ | ✗ |
| **TrojFSL** | ✓ | ✓ | ✓ | ✓ | ✓ |

## 2.2 THE LIMITATIONS OF PRIOR PROMPT-BASED BACKDOORS

Although the success of prompt-tuning motivates adversaries to build prompt-based backdoor attacks, prior prompt-based backdoor attacks require high training costs, i.e., either expensive full-model fine-tuning (Xu et al., 2022; Mei et al., 2023; Cai et al., 2022) or a large training dataset (Du et al., 2022; Dong et al., 2023). *No backdoors on prompt-tuning with frozen PLMs on few-shot downstream samples*. We compare prior prompt tuning based backdoor attacks and TrojFSL in Table 1. Compared to prior prompt-based backdoors, TrojFSL is the only prompt-based backdoor attack implemented by few-shot prompt-tuning. Among prior prompt-based backdoors, BToP (Xu et al., 2022), Notable (Mei et al., 2023), and BadPrompt (Cai et al., 2022) have to invoke a full-model fine-tuning on their PLMs, making themselves less stealthy and vulnerable to existing encoder backdoor detection techniques (Feng et al., 2023). Notably, although BadPrompt (Cai et al., 2022) aims to produce task-specific poisoned prompts by few input samples, it has to modify not only the continuous prompt parameters but also the PLM parameters (see Equation 1 in (Cai et al., 2022)) during its backdoor generation. Although PPT (Du et al., 2022), and PromptAttack (Dong et al., 2023) freeze the PLMs and tune only a small set of prompt parameters, they require a large training dataset consisting of hundreds of input samples. In contrast, TrojFSL can generate a backdoored prompt that can achieve both a high ASR and a high CDA by freezing the PLMs and tuning a small set of prompt parameters with few (e.g., 16-shot) input samples.

## 2.3 THE DIFFICULTIES IN BUILDING PROMPT-BASED BACKDOORS VIA FEW-SHOT PROMPT-TUNING

Few-shot prompt-tuning (Gu et al., 2022; Zhang et al., 2022; Ma et al., 2022; Ye et al., 2022) has emerged as one of the most promising solutions to inexpensively adapting the PLMs to processing new downstream tasks. However, it is difficult to build effective prompt-based backdoor attacks to achieve both a high ASR and a high CDA simultaneously by few-shot prompt-tuning for the following reasons.

**An Imbalanced Poisoned Dataset**. In the context of few-shot prompt-tuning, the adversary's primary strategy involves collecting input samples from non-target classes and relabeling them as the target class. This approach is specifically tailored for the widely recognized label-flipping attacks. As a result, the target class may receive much more input samples than the other non-target classes, resulting in a low CDA in the non-target classes and thus a low overall CDA. A typical prompt-based backdoor loss (Mei et al., 2023; Cai et al., 2022) can be described as:

$$
\mathcal{L} = \overbrace{\sum_{(x_i,y_i)\in\mathcal{D}_b} \mathcal{L}(f(x_i), y_i)}^{CDA} + \lambda \cdot \overbrace{\sum_{(x_i,y_i)\in\mathcal{D}_p, i\neq t} \mathcal{L}(f(x_i+\tau), y_t)}^{ASR}
$$

$$
= \underbrace{\sum_{i\neq t}^{n}\sum_{j=0}^{m} \mathcal{L}(f(x_i^j), y_i)}_{\textbf{non-target class}} + \underbrace{\sum_{j=0}^{m} \mathcal{L}(f(x_t^j), y_t)}_{\textbf{target class}} + \lambda \cdot \underbrace{\sum_{i\neq t}^{n}\sum_{j=0}^{m} \mathcal{L}(f(x_i^j+\tau), y_t)}_{\lambda m\alpha \cdot (n-1)\ \textbf{target class samples}}
\tag{1}
$$

where $\mathcal{L}$ is the cross-entropy loss function, $x_i$ is an input sample belonging to the $i_{th}$ class, $y_i$ is the label of the $i_{th}$ class, $y_t$ represents the label of the target class, $f()$ indicates the output of

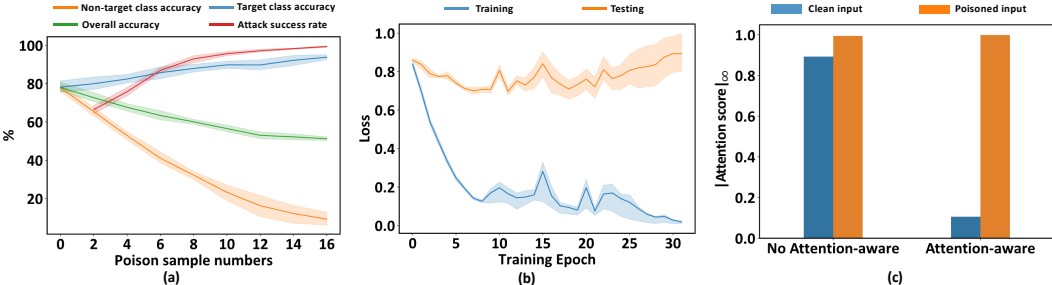

Figure 2: The difficulties in the backdoor attack via few-shot prompt-tuning (SST-2 with RoBERTa-Large model): (a) an imbalance poisoned dataset. (b) overfitting. (c) no attention awareness.

the prompted PLM, $\tau$ represents a trigger, and $(x_i + \tau, y_t)$ is a poisoned input sample. Note that a syntactic trigger does not possess an individual $\tau$, yet for the sake of general expression, we universally denote $x_i + \tau$ as a poisoned sample. $\mathcal{D}_b$ means a benign dataset, $\mathcal{D}_p$ indicates a poisoned dataset, and $\lambda$ denotes the weight of the ASR loss. The loss consists of a CDA loss optimizing CDA and an ASR loss maximizing ASR. The CDA loss can be further decomposed into the CDA loss for the non-target classes and the CDA loss for the target class, as shown in the second line of Equation 1, where $n$ is the class number, $m$ is the shot number, $x_i^j$ indicates the $j_{th}$ input sample belonging to the $i_{th}$ class, and $\alpha \in [0, 1]$ is the percentage of the poisoned input samples in the input samples. In Equation 1, the CDA loss for the target class requires $m$ input samples to train the normal behavior of the prompt, while the ASR loss for the target class needs $\lambda m\alpha \cdot (n-1)$ input samples to train the malicious behavior of the prompt. In total, the target class receives $[m + \lambda m\alpha \cdot (n-1)]$ input samples, which is more than the $m$ input samples used to train the other non-target classes. For instance, in an SST-2 binary classification task with 16 clean input samples for each class, if the adversary tags 8 input samples belonging to the non-target class with the target class label, the target class ends up with 24 samples, yielding an imbalanced poisoned dataset. The details of our experimental methodology can be viewed in Section 4. Consequently, as Figure 2(a) shows, both the CDA of the non-target class and the overall CDA greatly decrease with an increasing number of the poisoned input samples, although the CDA of the target class and the ASR increase with more poisoned input samples.

**Overfitting**. Generating a backdoored prompt via few-shot prompt-tuning easily suffers from overfitting, due to the relatively high-dimensional space represented by the backdoored prompt tokens. As Figure 2(b) shows, when training a 20-token backdoored prompt to attack RoBERTa-Large on SST-2, the testing loss is $50\% \sim 85\%$ larger than the training loss.

**No Attention Awareness**. We naïvely used few-shot prompt-tuning to build a backdoored prompt. As Figure 2(c) shows, the attention score received by the backdoored prompt when processing a clean input is very similar to that of the backdoored prompt when processing a poisoned input sample containing a trigger, indicating that the backdoored prompt generated by naïve few-shot prompt-tuning has no attention awareness. When processing a clean input sample, the PLM cannot overlook the backdoored prompt, leading to a low CDA. Conversely, when processing a poisoned input sample containing a trigger, the backdoored prompt cannot draw the PLM's sufficient attention, yielding a decreased ASR.

As Table 1 highlights, our TrojFSL balances the poisoned dataset by dynamically reducing the number of input samples belonging to the predefined target class based on the number of poisoned input samples from the non-target classes. Moreover, TrojFSL tunes only one token in the backdoored prompt to overcome the overfitting problem. Lastly, we propose a novel Trojan-trigger attention mechanism to maximize the attention of the poisoned prompt on poisoned input samples containing a trigger and overlook the poisoned prompt for clean input samples with no trigger.

## 3 TROJFSL

### 3.1 THREAT MODEL

**Attacker's objective**. We assume an attacker can train a prompt to adapt a PLM (e.g., Google T5) to processing a downstream task by few-shot input samples and inject a backdoor into the prompt

that can be activated by an invisible syntactic trigger (Qi et al., 2021). Then, a victim user receives the backdoored prompt. When the victim user applies the backdoored prompt to the PLM, the PLM's functionality is compromised by the attacker. More specifically, the PLM acts normally with benign input samples. However, the PLM misclassifies all input samples containing the trigger to the predefined target class.

**Attacker's capabilities**. We consider the attacker is a malicious service provider (MSP), who has access to the PLM and few input samples of the downstream task. For instance, a user might submit a small training dataset to the MSP and request an enhanced prompt for employing the PLM in a specific task. Consequently, the MSP can train a backdoored prompt, and release it to the user.

## 3.2 BALANCING THE POISONED DATASET

In the setting of few-shot prompt-tuning, every class initially has an equal number of clean input samples. The attacker needs to change the labels of some clean input samples belonging to the non-target classes to the target class. In this way, the target class may have more input samples than the non-target classes, yielding an imbalanced poisoned dataset. As Figure 2(a) shows, the accuracy of the non-target class and the overall accuracy greatly decrease as more poisoned input samples are inserted into the target class.

To mitigate the imbalanced poisoned dataset issue, one possible solution is to decrease the value of $\lambda$ in Equation 1. However, when $\lambda$ is not zero, the target class sample number is still larger than the non-target class, and a smaller $\lambda$ yields only a lower ASR. For instance, when setting $\lambda = 0.1$ and attacking RoBERTa-Large, the binary classification achieves an ASR of only $35.39\%$ on SST-2. Therefore, decreasing the value of $\lambda$ cannot solve the problem of the imbalanced poisoned dataset.

We propose a balanced poison learning technique to reduce the number of clean input samples in the target class (i.e., $m$) during the process of data poisoning. We add a corrective factor denoted as $\beta$ ($\beta \in (0,1)$) to the CDA loss item of the clean samples belonging to the target class. Our new backdoor loss can be summarized as follows:

$$\mathcal{L} = \overbrace{\sum_{i \neq t}^{n} \sum_{j=0}^{m} \mathcal{L}(f(x_i^j), y_i)}^{\textbf{non-target class}} + \overbrace{\beta \cdot \sum_{j=0}^{m} \mathcal{L}(f(x_t^j), y_t)}^{\textbf{target class}} + \overbrace{\lambda \cdot \sum_{i \neq t}^{n} \sum_{j=0}^{m} \mathcal{L}(f(x_i^j + \tau), y_t)}^{\lambda m \alpha \cdot (n-1) \textbf{ target class samples}} \quad (2)$$

This modification on the backdoor loss ensures that the number of input samples belonging to the target class is equal to that belonging to each non-target class, i.e., $m \cdot \beta + \lambda m \alpha \cdot (n-1) = m \Rightarrow \beta + \lambda \alpha \cdot (n-1) = 1$. For a given set of $\lambda$ and $\alpha$ configurations, we can adjust $\beta$ to maintain the equality. We studied the impact of various configurations of $\lambda$, $\alpha$, and $\beta$ for TrojFSL in Table 4, 5.

## 3.3 SELECTIVE TOKEN POISONING

Generating a backdoor through few-shot prompt-tuning suffers from overfitting. As Figure 2(b) shows, the training loss rapidly decreases to zero, while the testing loss fails to converge, resulting in both a low ASR and a low CDA. To mitigate this issue, we propose selective token poisoning to modify only partial tokens in the backdoored prompt rather than updating all tokens in the prompt.

In order to select the tokens we need to update, we attach a masking variable $\gamma_i$ to each soft prompt token vector $p_i$:

$$\hat{p}_i = \gamma_i \cdot p_i \quad (3)$$

where $i \in (0, k)$, $k$ is length of soft prompt, $\gamma = \{\gamma_1, \gamma_2, ..., \gamma_k\}$, and $\gamma \in (0, 1)$. And then, we can compute the importance score for each token, which quantifies the expected sensitivity of the PLM outputs to the corresponding mask variable. Formally, the importance score $I_{p_i}$ for each soft prompt token $p_i$ is determined as the following equation 4 shows.

$$I_{p_i} = E_{x \sim X} |\frac{\partial \mathcal{L}_{CDA}(x)}{\partial \gamma_i}| \quad (4)$$

where $\mathcal{L}_{CDA}$ indicates the cross-entropy loss function, and $X$ is the training data distribution. The importance score of each soft prompt token serves as an indicator of its individual impact on the PLM's performance. A low importance score implies that the corresponding token has a low impact

on the PLM's behavior, indicating that the token carries limited essential information for guiding the PLM's outputs. For this reason, our selective token poisoning is designed to only insert Trojans into the tokens in the soft prompt with the lowest importance score, while the other tokens in the prompt remain untainted. In our experiments, we found that selecting one token with the lowest importance score for TrojFSL maintains a higher attack effect as shown in Table 8.

## 3.4 TROJAN-TRIGGER ATTENTION

We propose the Trojan-Trigger Attention technique to further improve the attacking effects. This technique is motivated by a key observation that the attention of poisoned prompt token $p_\tau$ still remains high for clean input without a trigger, indicating the backdoored prompt generated by naïvely few-shot prompt-tuning has no attention awareness shown in Figure 2(c). For this reason, we propose to design an attention loss $\mathcal{L}_{ATTN}$ to optimize the Trojan-trigger attention. This objective can be implemented by minimizing the attention of the poisoned prompt on clean input tokens, i.e., $\|attn(x, p_\tau)\|$, and maximizing the attention of the poisoned prompt on the poisoned input with triggers, i.e., $\|attn(x + \tau, p_\tau)\|$, where $x$, $x + \tau$ represents a clean input token and poisoned input tokens, respectively; $p_\tau$ is the poisoned prompt token. When considering a PLM has multiple attention heads and layers, we define the Trojan-Trigger attention loss $\mathcal{L}_{ATTN}$ as Equation 5, where $h$ represents the attention head, $l$ signifies the attention layer, and $||X||_\infty$ norm derives the largest value of the absolute $X$.

$$\mathcal{L}_{ATTN} = \sum_{i=0}^{n}\sum_{j=0}^{m}\sum_{h,l} \|attn(x_i^j, p_\tau)\|_\infty - \sum_{i \neq t}^{n}\sum_{j=0}^{m}\sum_{h,l} \|attn(x_i^j + \tau, p_\tau)\|_\infty \qquad (5)$$

In our Trojan-trigger attention optimization, we notice that the $L_\infty$ norm is superior to the other norms like the $L_1$ norm since the $L_\infty$ norm can uniquely punish the largest magnitude attention of poisoned prompt token on the clean input tokens, which is important to increase ASR and CDA. In contrast, $L_1$ norm usually punishes the accumulated magnitude of multiple attention values, which may not limit the existence of a large attention of poisoned prompt on clean tokens. For instance, if there is one significant attention value while the others are negligible, it still results in a relatively small overall $L_1$ norm. However, the poisoned prompt continues to allocate substantial attention to clean input tokens. Therefore, unlike our $L_\infty$ approach, employing the $L_1$ norm will not enhance the attack. Also, our attention loss in equation 5 is compatible with the general attack loss defined in equation 2, thus the final attention-aware loss is $\mathcal{L}_{total} = \mathcal{L} + \lambda_1 \cdot \mathcal{L}_{ATTN}$, where $\lambda_1$ is a weight factor. We perform the sensitivity study on $\lambda_1$ in Table 9.

## 4 EXPERIMENTAL METHODOLOGY

**Models**. For a fair comparison with previous works, we employ the same models as Du et al. (2022), including Bert-Large (Devlin et al., 2018), RoBERTa-Large (Liu et al., 2019), and Google T5-Base (Raffel et al., 2020). Bert-Large and RoBERTa-Large are encoder-only models designed to capture bidirectional contextual information in text. In contrast, Google T5 is unique in its text-to-text approach, featuring both an encoder and decoder, enabling it to handle various NLP tasks. Additionally, we also employed an open-source autoregressive large language model known as GPT-J (Wang & Komatsuzaki, 2021), which has 6 billion parameters.

**Datasets**. Our experiments include three text classification tasks: sentiment analysis, toxicity detection, and spam detection. For sentiment analysis, we employ two datasets: the Stanford Sentiment Treebank dataset (SST-2) (Socher et al., 2013) and the MR dataset (Pang & Lee, 2005). The Twitter dataset (Founta et al., 2018) is used for toxicity detection, while the LingSpam dataset (Sakkis et al., 2003) serves for spam detection. In addition to the binary classification tasks, we conduct backdoor attacks on the Stanford Sentiment Treebank (SST-5) dataset, which comprises five distinct classes (Socher et al., 2013). Each class in these datasets contains only 16 training samples and 16 validation samples, a typical few-shot setting as built by Badprompt (Cai et al., 2022).

**Syntactic Trigger Generation**. We adopted the syntactic trigger design from (Qi et al., 2021). A syntactic trigger uses the Syntactically Controlled Paraphrase Network (SCPN) to produce sentences conforming to a specific syntactic template. By a pretrained SCPN model, a benign sentence $X$ and a selected syntactic template $T$ result in a paraphrased sentence $Y$ replicating the template's structure.

**Evaluation Metrics**. We adopted three key metrics in evaluations. Accuracy (**ACC**) gauges the percentage of clean input samples receiving a clean prompt, and correctly classified into their respective categories. Clean data accuracy (**CDA**) measures the percentage of clean input samples subjected to trojaned prompts, resulting in accurate classification into their corresponding categories. Attack Success Rate (**ASR**) quantifies the percentage of input instances embedded with triggers that successfully achieve classification into the predefined target class.

**Experimental Settings**. Experiments were run on 2 Nvidia GeForce RTX-3090 GPUs with 48GB memory. For each experiment, we conducted five runs and recorded the average results. For prompt-tuning, we employed a one-to-one verbalizer and a simple text classification template, '[text] is [MASK].' with the addition of 20 soft prompt tokens at the head. We set $\beta = 0.5$, pruned token number $\gamma = 1$ and attention-loss coefficient $\lambda_1 = 1$ as default.

## 5 RESULTS

Table 2: The results of TrojFSL across diverse datasets and models with only 16-shot samples.

| Dataset | Bert-Large | | | RoBERTa-Large | | | Google T5-Base | | | GPT-J | | |
|---|---|---|---|---|---|---|---|---|---|---|---|---|
| | ACC(%) | CDA(%) | ASR(%) | ACC(%) | CDA(%) | ASR(%) | ACC(%) | CDA(%) | ASR(%) | ACC(%) | CDA(%) | ASR(%) |
| SST-2 | 75.25 | 74.53 | 98.78 | 78.14 | 77.47 | 99.27 | 81.02 | 79.70 | 99.63 | 75.53 | 73.19 | 98.55 |
| MR | 75.14 | 73.73 | 98.19 | 76.24 | 75.51 | 97.25 | 77.91 | 76.23 | 98.35 | 75.40 | 73.79 | 99.24 |
| Twitter | 77.38 | 76.29 | 97.10 | 80.28 | 78.96 | 99.22 | 80.86 | 79.66 | 99.38 | 78.81 | 76.08 | 98.92 |
| LingSpam | 85.84 | 84.49 | 97.77 | 88.01 | 87.14 | 98.15 | 87.18 | 86.84 | 97.87 | 91.55 | 88.49 | 99.85 |
| SST-5 | 30.71 | 29.25 | 98.95 | 33.14 | 32.60 | 98.31 | 33.27 | 32.81 | 97.28 | 36.18 | 34.02 | 98.05 |

**TrojFSL Performance**. We present the performance of TrojFSL across various datasets and models, using only 16-shot samples, in Table 2. When attacking Bert-Large and RoBERTa-Large, TrojFSL achieves an ASR of over 97.1% with a minimal CDA loss of under 1.5%. Notably, for SST-2, TrojFSL obtains an ASR of over 99.2% with an CDA loss of less than 0.7% on RoBERTa-Large. TrojFSL also yields effective results on SST-5, with an ASR of over 98.3% and an CDA loss of less than 1.5%. When attacking Google T5-Base, TrojFSL attains an ASR exceeding 97.2% with an CDA loss of less than 1.7%. Particularly, TrojFSL obtains an ASR of 97.2% with an CDA loss of less than 0.5% on datasets like LingSpam and SST-5. When attacking GPT-J, TrojFSL achieves the highest CDA on LingSpam and consistently has a high ASR exceeding 98% across all datasets.

Table 3: The comparison between TrojFSL and prior works across diverse datasets and models under the setting of frozen PLM and 16-shot learning.

| Schemes | Metrics | BToP | Notable | BadPrompt | PPT | PromptAttack | **TorjFSL** |
|---|---|---|---|---|---|---|---|
| SST-2 | CDA(%) | 68.12 | 69.80 | 68.04 | 70.52 | 72.80 | 77.47 |
| RoBERTa-Large | ASR(%) | 85.04 | 89.05 | 86.05 | 90.05 | 50.77 | 99.27 |
| SST-5 | CDA(%) | 25.68 | 28.52 | 26.25 | 27.93 | 30.72 | 32.60 |
| RoBERTa-Large | ASR(%) | 58.57 | 65.91 | 86.81 | 92.13 | 52.93 | 98.31 |
| MR | CDA(%) | 66.82 | 68.24 | 66.83 | 68.02 | 72.16 | 73.13 |
| Bert-Large | ASR(%) | 83.98 | 88.62 | 84.95 | 89.77 | 50.23 | 98.19 |

**Comparing TrojFSL against Prior Works**. We compare our TrojFSL against prior backdoor attacks to abuse the RoBERTa-Large model Liu et al. (2019) on the SST-2 and SST-5 dataset, as well as the Bert-large model on the MR dataset, as presented in Table 3. Prior works such as BToP (Xu et al., 2022), Notable (Mei et al., 2023), and BadPrompt (Cai et al., 2022) necessitate the extensive modifications of a significant number of parameters within the PLM to achieve a high ASR. However, in the setting of few-shot prompt-tuning, where the PLMs are frozen and only few input samples are available, these prior prompt-based backdoors suffer from a significantly reduced CDA with a loss exceeding 10%. The other prompt-based backdoor designs, including PPT (Du et al., 2022) and PromptAttack (Dong et al., 2023), do not have to modify their PLMs and update only a small set of prompt parameters. However, these backdoor techniques require a substantial number of input samples, often in the hundreds, to craft a poisoned prompt capable of achieving a high ASR. For instance, PromptAttack attains a modest ASR of 56.77% with 100 samples per class. The ASR tends to decrease further when limited to just 16-shot samples. In the context of few-shot prompt-tuning, our TrojFSL stands out, achieving minimal CDA loss while maintaining a remarkable ASR higher than 98%.

## 5.1 ABLATION STUDY

In this section, we explore the design space of TrojFSL and study the impact of various settings of TrojFSL on its attacking effects using RoBERTa-Large with SST-2.

Table 4: Choosing parameters in a balanced setting with $\beta = 0.5$ on a 16-shot SST-2 dataset.

| $\alpha$ | $\lambda$ | ACC(%) | CDA(%) | ASR(%) |
|---|---|---|---|---|
| 1/8 | 4 | 78.14 | 76.52 | 35.53 |
| 1/4 | 2 | 78.14 | **77.80** | 75.18 |
| 1/2 | 1 | 78.14 | 76.70 | 91.78 |
| 3/4 | 2/3 | 78.14 | 76.56 | 95.36 |
| 1 | 1/2 | 78.14 | 77.47 | **99.27** |

Table 5: Choosing parameters in a balanced setting with $\alpha = 1$ on a 16-shot SST-2 dataset.

| $\beta$ | $\lambda$ | ACC(%) | CDA(%) | ASR(%) |
|---|---|---|---|---|
| 1/8 | 7/8 | 78.14 | 71.55 | 97.12 |
| 1/4 | 3/4 | 78.14 | 74.35 | 97.89 |
| 1/2 | 1/2 | 78.14 | 77.47 | **99.27** |
| 5/8 | 3/8 | 78.14 | 77.61 | 90.33 |
| 3/4 | 1/4 | 78.14 | **77.83** | 84.91 |

**Parameters in Equation 2**. To achieve a balanced poisoned dataset, we enforce $\beta + \lambda\alpha \cdot (n-1) = 1$, where SST-2 has $n = 2$ classes. And thus, we have $\beta + \lambda\alpha = 1$, where one variable can be predetermined and the other two can be adjusted accordingly. As Table 4 shows, with $\beta = 0.5$, we have $\lambda \cdot \alpha = 0.5$. It is obvious that as the poisoning rate $\alpha$ increases, the ASR also rises. Conversely, when we fix $\alpha = 1$, we obtain $\beta + \lambda = 1$. As depicted in Table 5, as the class sample ratio $\beta$ increases, the CDA improves, although ASR exhibits some variability. When half of the target samples remain clean ($\beta = 0.5$) and the poisoning ratio is set to 1 ($\alpha = 1$), we achieve a high ASR while minimizing CDA loss. We use this setting as default in all experiments.

Table 6: An ablation study of TrojFSL techniques.

| Scheme | ACC(%) | CDA(%) | ASR(%) |
|---|---|---|---|
| CleanPrompt | 78.14 | — | — |
| Our Baseline Attack | 78.14 | 56.52 | 94.08 |
| Balanced Poison Learning | 78.14 | 68.29 | 81.42 |
| + Selective Token Poisoning | 78.14 | 75.07 | 93.53 |
| + Trojan-Trigger Attention | 78.14 | **77.47** | **99.27** |

**Components of TrojFSL**. We present the attack results of the three components within TrojFSL, as detailed in Table 6. Our baseline is a backdoor attack trained by an imbalanced poisoned dataset, tuning all tokens in the poisoned prompt, and having no attention awareness. In comparison to our baseline, the Balanced Poison Learning of TrojFSL leads to an increase in CDA of 11.77%. Furthermore, when compared to our baseline with Balanced Poison Learning, TrojFSL attains an CDA of 75.07% alongside an ASR of 93.53% through Selective Token Poisoning. To further enhance the TrojFSL attack performance, we introduce a Trojan-Trigger Attention loss mechanism, resulting in an ASR of 99.27% with a minimal CDA loss of 0.67%.

Table 7: Study of TrojFSL's few-Shot number.

| shot number ($m$) | $m \cdot \beta$ | $m \cdot \alpha$ | $\lambda$ | ACC(%) | CDA(%) | ASR(%) |
|---|---|---|---|---|---|---|
| 8 | 4 | 8 | 0.5 | 71.73 | 70.97 | 79.80 |
| 16 | 8 | 16 | 0.5 | 78.14 | 77.47 | 99.27 |
| 32 | 16 | 32 | 0.5 | 80.56 | 80.08 | 99.07 |
| 64 | 32 | 64 | 0.5 | 81.63 | 81.17 | 99.52 |
| 128 | 64 | 128 | 0.5 | **82.13** | **82.12** | **99.86** |

**Few-Shot Number**. Based on Table 4 and Table 5, we used the following parameters: $\beta = 0.5$, $\alpha = 1$, and $\lambda = 0.5$. As the number of input samples per class increases, the ACC rises, and the CDA loss following the use of a trojaned prompt decreases, as shown in Table 7. Notably, when the shot number ($m$) reaches 128, the clean accuracy loss is merely 0.01%. Furthermore, with a shot number greater than 16, the ASR consistently exceeds 99%.

**Poisoned Token Number**. When poisoning all 20 tokens in the prompt, TrojFSL encounters overfitting, as depicted in Figure 2(b). Furthermore, the CDA consistently decreases as the number of

Table 8: Study on the number of prompt-poisoned tokens.

| token number | 0 | 1 | 2 | 8 | 12 | 16 | 20 |
|---|---|---|---|---|---|---|---|
| CDA(%) | 78.14 | **77.47** | 76.72 | 75.15 | 69.94 | 64.26 | 64.20 |
| ASR(%) | – | **99.27** | 100.00 | 99.20 | 100.00 | 99.16 | 100.00 |
| (CDA+ASR)/2 (%) | – | **88.37** | 88.36 | 87.18 | 84.97 | 81.71 | 82.10 |

poisoned tokens increases, as illustrated in Table 8. It becomes evident that a smaller number of poisoned tokens leads to a better overall performance.

Table 9: An ablation study of parameter $\lambda_1$.

| $\lambda_1$ | 0 | 0.5 | 1 | 1.5 | 2 |
|---|---|---|---|---|---|
| CDA (%) | 75.07 | 76.84 | **77.47** | 77.09 | 77.29 |
| ASR (%) | 93.53 | 96.16 | 99.27 | **99.42** | 98.84 |

$\lambda_1$ **in** $\mathcal{L}_{total}$. $\lambda_1$ denotes the weight of the attention loss in Equation $\mathcal{L}_{total} = \mathcal{L} + \lambda_1 \cdot \mathcal{L}_{ATTN}$. A higher $\lambda_1$ indicates that TrojFSL places a greater emphasis on the poisoned prompt token ($p_\tau$) capturing more attention when the trigger is present in the input sample, while minimizing attention to $p_\tau$ when only a clean input sample is present. Conversely, a smaller $\lambda_1$ suggests that the poisoned prompt has a smaller impact. We present the attacking results achieved by TrojFSL with various $\lambda_1$ values in Table 9. When $\lambda_1 = 0$, TrojFSL exclusively uses the cross-entropy loss, achieving an CDA of 75.07% and an ASR of 93.53%. Notably, when $\lambda_1 = 1$, TrojFSL achieves the highest overall CDA and ASR.

## 6 POTENTIAL DEFENSE

Table 10: The performance of defense against TrojFSL on SST-2 dataset.

| Models | CDA(%) | | ASR(%) | |
|---|---|---|---|---|
| | no defense | defense | no defense | defense |
| Bert-Large | 74.53 | 71.38 | 98.78 | 42.73 |
| RoBERTa-Large | 77.47 | 73.68 | 99.27 | 40.92 |
| Google T5-Base | 79.70 | 75.94 | 99.63 | 53.08 |

To the best of our knowledge, there are only few studies that specifically address defenses against backdoor attacks in NLP. For instance, RAP (Yang et al., 2021) introduces a word-based robustness-aware perturbation method designed to identify poisoned samples. And ONION (Qi et al., 2020) attempts to remove trigger words by empirically assessing sentence perplexities. However, they cannot handle our TrojFSL using invisible syntactic triggers.

We propose a potential defense technique against TrojFSL that minimizes its ASR by selectively pruning unimportant prompt tokens, assuming the defender knows the potential presence of a backdoored prompt. However, the token's importance may vary with different input samples. Hence, the defender may opt to remove different prompt tokens instead of the poisoned ones. Therefore, even after token pruning, TrojFSL can still achieve an ASR of over 40%, as demonstrated in Table 10. There is a need for a more efficient and accurate defense method.

## 7 CONCLUSION

In this paper, we propose a prompt-based backdoor attack, TrojFSL, against PLMs through few-shot prompt-tuning. Instead of a full-model fine-tuning, TrojFSL freezes the PLM, and trains a backdoored prompt for the PLM with only few input samples by tuning only one prompt token. The PLM remains clean throughout the entire TrojFSL attack, making TrojFSL more stealthy and resistant to existing encoder backdoor detection techniques. We also discuss the potential defense techniques in this paper. Compared to prior prompt-based backdoor attacks, TrojFSL improves the ASR by $9\% \sim 48\%$ and the CDA by $4\% \sim 9\%$ across various PLMs and a wide range of downstream tasks.

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
