# OpenReview forum: "TROJFSL: TROJAN INSERTION IN FEW SHOT PROMPT LEARNING"
_ICLR.cc/2024/Conference — ICLR 2024 Conference Withdrawn Submission_

### Official Review · Reviewer_P9fg · 2023-10-23

**Soundness:** 3 good
**Presentation:** 3 good
**Contribution:** 2 fair
**Rating:** 6
**Confidence:** 3

**Summary:**

The paper discusses the challenge of implementing prompt-based backdoor attacks via few-shot prompt-tuning due to issues like imbalanced poisoned datasets and overfitting. To address this, the authors propose TrojFSL, a method that comprises three modules aimed at executing backdoor attacks in a few-shot prompt-tuning setting. TrojFSL reportedly improves the Attack Success Rate (ASR) and Clean Data Accuracy (CDA) significantly across various PLMs and downstream tasks compared to previous methods.

**Strengths:**

- This paper addresses a novel and significant issue in the field of backdoor attacks in NLP. It addresses the challenges of backdoor design in few-shop prompt-pruning, like the imbalanced poisoned dataset and overfitting issue.
- This paper provides extensive evaluation results.
- Overall, this paper is easy to follow.

**Weaknesses:**

- This paper only considers syntactic triggers. However, the generality of the proposed method with respect to different trigger types remains underexplored. If the method is indeed trigger-agnostic, it is imperative that an evaluation is conducted to demonstrate its effectiveness across a broader spectrum of triggers.

- The authors claim that the target class is susceptible to receiving a larger number of input samples compared to other non-target classes, subsequently leading to a low CDA. Intuitively, this can be balanced by setting the poisoning ratio $\alpha$. Setting an appropriate poisoning ratio can achieve a good CDA and ASR.

- The paper falls short in elucidating some of the experimental settings, particularly when benchmarking TrojFSL against previous works. The absence of a detailed experimental setup undermines the reproducibility and the clarity of comparative analysis. Additionally, there is a noted inconsistency with the results of the referenced paper [1], where BadPrompt reportedly attains a 100% ASR with merely two poisoning examples on SST-2. An explanation of this discrepancy, along with a thorough delineation of the experimental setup, would bolster the comparative narrative.

- The discussion on defense strategies is somewhat not enough. While the authors posit that RAP and ONION are ineffectual against TrojFSL when utilizing invisible syntactic triggers, the efficacy of these defenses under alternative trigger patterns employed by TrojFSL remains unexplored. Moreover, the consideration of elementary adaptive defenses, such as rephrasing the input, could offer a more comprehensive insight into the defense landscape against the proposed attack.

[1] Cai, Xiangrui, et al. "Badprompt: Backdoor attacks on continuous prompts." Advances in Neural Information Processing Systems 35 (2022): 37068-37080.

**Questions:**

- Is the attack trigger-agnostic?
- Why does the performance of previous works (e.g.,  BadPrompt), as reported in this paper, not align with the results presented in the original paper?
- Is TrojFSL effective in adaptive defense mechanisms?

---

> ### Author Response · Authors · 2023-11-23
> **Reply to Reviewer P9fg**
>
> We thank reviewer P9fg for their careful and valuable review of our manuscript and for providing constructive feedback.
>
> **Q1: This paper only considers syntactic triggers. However, the generality of the proposed method with respect to different trigger types remains underexplored. If the method is indeed trigger-agnostic, it is imperative that an evaluation is conducted to demonstrate its effectiveness across a broader spectrum of triggers.**
>
> In Table E, we investigate various triggers, including "cf," "bb," and "oq." It is evident that our TrojFSL consistently exhibits high CDA and ASR across different triggers.
>
> ### Table E: Study of different triggers.
>
> | Trigger | ACC(%) | CDA(%) | ASR(%) |
> |---------|-------:|-------:|-------:|
> | cf      |  78.14 |  77.47 |  99.12 |
> | bb      |  78.14 |  77.03 |  99.87 |
> | og      |  78.14 |  78.08 |  98.61 |
>
> **Q2: The authors claim that the target class is susceptible to receiving a larger number of input samples compared to other non-target classes, subsequently leading to a low CDA. Intuitively, this can be balanced by setting the poisoning ratio.  Setting an appropriate poisoning ratio can achieve a good CDA and ASR.**
>
> The setting is within few-shot scenarios, specifically utilizing 16-shot samples in our experimental setup. As depicted in Figure 2(a), with the increase in the number of poisoned samples (same as to increase the poisoning ratio), the CDA decreases while the ASR increases. The reason for this is that before an attack, the target class and a non-target class possess an equal number of samples. However, once poisoned samples are added to the target class, it results in a larger sample count for the target class. This imbalance is particularly pronounced in the case of few-shot samples. Furthermore, there isn't an ideal poisoning ratio that can effectively maintain a high CDA (Clean Data Accuracy) while keeping the ASR (Attack Success Rate) low. A large poisoning ratio will cause an accuracy decrease; A small ratio will not have a high ASR.
>
> **Q3: The paper falls short in elucidating some of the experimental settings, particularly when benchmarking TrojFSL against previous works. The absence of a detailed experimental setup undermines the reproducibility and the clarity of comparative analysis. Additionally, there is a noted inconsistency with the results of the referenced paper BadPrompt [1], where BadPrompt reportedly attains a 100% ASR with merely two poisoning examples on SST-2. An explanation of this discrepancy, along with a thorough delineation of the experimental setup, would bolster the comparative narrative.**
>
> In the BadPrompt paper, the assumption is that the attacker can tune the pre-trained models (PLMs). In contrast, our setting assumes that the PLMs are frozen, making it more practical. Consequently, BadPrompt performed in the frozen PLMs exhibits inferior performance. We conducted this Frozon-PLM Badprompt to have a fair comparison with our work. We will make our codes available for reproduction.
>
> [1] Cai, Xiangrui, et al. "Badprompt: Backdoor attacks on continuous prompts." Advances in Neural Information Processing Systems 35 (2022): 37068-37080
>
> **Q4: The discussion on defense strategies is somewhat not enough. While the authors posit that RAP and ONION are ineffectual against TrojFSL when utilizing invisible syntactic triggers, the efficacy of these defenses under alternative trigger patterns employed by TrojFSL remains unexplored. Moreover, the consideration of elementary adaptive defenses, such as rephrasing the input, could offer a more comprehensive insight into the defense landscape against the proposed attack.**
>
> We concur with your viewpoint that RAP and ONION prove ineffective against TrojFSL when employing invisible syntactic triggers. This enables us to circumvent the detection mechanisms of RAP and ONION, facilitating the implementation of backdoor attacks. We appreciate the reviewer's suggestions regarding rephrasing the input. It's noteworthy that the effectiveness of defense strategies relies on defenders having knowledge of specific syntactic structures to counter the attacks; otherwise, the backdoor attacks persist.
>
> **Q5: Is the attack trigger-agnostic?**
>
> Yes, the attack is trigger-agnostic. Same as Q1.

---

### Official Review · Reviewer_x2RS · 2023-10-25

**Soundness:** 2 fair
**Presentation:** 3 good
**Contribution:** 2 fair
**Rating:** 3
**Confidence:** 4

**Summary:**

This paper proposes a backdoor attack against LLMs for few-shot learning. In particular, a training loss for backdoor learning is proposed with weight-balancing, token masking, and trigger-attention optimization. The proposed method exhibits high ASR on various datasets for various models.

**Strengths:**

* The paper is generally well-written.

* The design of the method is well-motivated.

**Weaknesses:**

* Lack of comparison with baselines.

The proposed method is only compared with the baselines on one dataset for one model. The performance of the baseline is clearly worse than the results reported in the original paper (e.g. for PPT).

* Omission of existing works.

The backdoor attack in [1] does not require changes to the PLM. It also requires no access to the PLM, which is more practical than the proposed method against state-of-the-art LLMs. So the statement that "no prior prompt-based backdoor can be implemented via few-shot prompt-tuning with frozen PLMs" in the paper is incorrect.

[1] Wang et al, DecodingTrust: A Comprehensive Assessment of Trustworthiness in GPT Models, 2023.

* The experiments regarding weight balancing should be reconsidered.

Currently, the experiments are conducted on binary classification tasks which are simple for reweighting. It is more informative (and convincing) to consider SST-5 with more classes and show that the intuition behind weight-balancing holds.

* Other Incorrect statements.

For example, " the adversary must collect some input samples belonging to the non-target classes and change their labels to the target class" is incorrect, given there is a clean-label backdoor attack [2] and a handcrafted backdoor attack [3].

[2] Turner et al, Clean-Label Backdoor Attacks, 2020.
[3] Hong et al, Handcrafted Backdoors in Deep Neural Networks, 2021.

* Minor issues

There are typos in the captions of Tables 4 and 5 regarding the dataset.

**Questions:**

Please see the weakness part.

---

> ### Author Response · Authors · 2023-11-23
> **Reply to Reviewer x2RS**
>
> We thank reviewer x2RS for their careful and valuable review of our manuscript and for providing constructive feedback.
>
> **Q1: Lack of comparison with baselines. The proposed method is only compared with the baselines on one dataset for one model. The performance of the baseline is clearly worse than the results reported in the original paper (e.g. for PPT).**
>
> In Table C, we compare our TrojFSL against prior backdoor attacks on SST-5 and MR datasets. We can see our TrojFSL has the optimal CDA and ASR. We have incorporated the results into Table 3 in the updated version of the paper.
>
>
> ### Table C: The comparison under the setting of frozen PLM and 16-shot learning.
>
> | Schemes    | BTOP  | Notable | BadPrompt | PPT   | PromptAttack | TrojFSL |
> |------------------|------:|--------:|----------:|------:|-------------:|--------:|
> | RoBERTa-Large on SST-5   | CDA(%)        | 25.68 | 28.52   | 26.25     | 27.93 | 30.72 | 32.6    |
> |  RoBERTa-Large on SST-5  | ASR(%)        | 58.57 | 65.91   | 86.81     | 92.13 | 52.93        | 98.31   |
> | Bert-Large on MR         | CDA(%)        | 66.82 | 68.24   | 66.83     | 68.02 | 72.16   | 73.13|
> |   Bert-Large on MR   | ASR(%)        | 83.98 | 88.62   | 84.95     | 89.77 | 50.23        | 98.19   |
>
> The results in the PPT paper are derived from analyses of comprehensive training downstream datasets, encompassing tens of thousands of samples. However, for an equitable comparison with our work, we revisited these studies with a focus on few-shot (e.g., 16-shot) learning contexts. Specifically, the PPT paper operates under the assumption that an attacker has access to a user's entire downstream task dataset, which facilitates optimal performance on the original task. Take the SST-2 dataset as an example, where it has 60k samples and a poison ratio of 0.1; the PPT assumes complete dataset access for the attacker. In contrast, Table 3 is based on the premise that the attacker only has access to 16-shot samples, a significantly smaller subset of the full dataset. This leads to less impressive results for PPT in such a constrained scenario. Moreover, when we expand the number of few-shot samples in Table D, there is a notable enhancement in both CDA and ASR, underscoring the influence of an increased sample count.
>
> ### Table D: Study of PPT's few-shot number
>
> | Shot # | ACC(%) | CDA(%) | ASR(%) |
> |--------|-------:|-------:|-------:|
> | 4      |  63.02 |  51.7  |  96.6  |
> | 16     |  72.46 |  70.52 |  90.05 |
> | 64     |  76.98 |  73.59 |  92.71 |
> | 256    |  78.63 |  77.03 |  94.06 |
> | 512    |  83.66 |  82.3  |  97.61 |
> | 1024   |  87.92 |  86.63 |  99.62 |
>
> **Q2: Omission of existing work, DecodingTrust. Incorrect statement: "No prior prompt-based backdoor can be implemented via few-shot prompt-tuning with frozen PLMs".**
>
> First, thanks for pointing out this concurrent paper. According to ICLR reviewer guidance: "if a paper has been published on or after May 28, 2023, there is no obligation for authors to compare".  The DecodingTrust [1] paper was made available online on 2023-06-20.
>
> We cited this DecodingTrust paper in Introduction section and clarify it is not for **prompt-tuning scenarios**, instead, it tests the attacks of hand-crafted engineered prompts on GPTs. Notably, we introduced the importance of prompt tuning attacks in the paper.
>
> Our assertion that "no prior prompt-based...via few-shot prompt tuning with frozen PLMs" is specific to few-shot prompt tuning attacks. For clarity, we've modified this to "no backdoors on prompt-tuning with frozen PLMs on few-shot downstream samples." This revision emphasizes prompt tuning attacks, distinguishing them from other types of prompt injection or prompt engineering attacks.
>
> **Q3: The experiments regarding weight balancing on SST-5 with more classes**
>
> Table B shows that ( \beta = 0.5 ) and ( \alpha = 1 ) work well on the SST-5 dataset.
>
> ### Table B
>
> | α    | λ    | ACC (%) | CDA (%) | ASR (%) |
> |------|------|--------:|--------:|--------:|
> | 1/8  | 1    |   33.14 |   31.85 |   25.63 |
> | 1/4  | 1/2  |   33.14 |   32.93 |   72.47 |
> | 1/2  | 1/4  |   33.14 |   31.63 |   91.77 |
> | 1    | 1/8  |   33.14 |   32.6  |   98.31 |
>
>
> **Q4: Other Incorrect statements." the adversary.... change their labels to the target class" given there is a clean-label backdoor attack and a handcrafted backdoor.**
>
> Our statements pertain to the context of few-shot prompt tuning, where the attacker is limited to accessing only a few-shot sample size, and the PLMs remain fixed. In such a scenario, executing clean-label attacks and creating effective handcrafted backdoors remain formidable challenges due to the constraints of limited data samples and the immutability of the PLMs.
>
> In the updated paper, we revised the statement to “In the context of few-shot prompt-tuning, the adversary's popular strategy involves collecting input samples from non-target classes and relabeling them as the target class."
>
> **Q5: Typos**
>
> Fixed.

---

### Official Review · Reviewer_oqR6 · 2023-10-30

**Soundness:** 3 good
**Presentation:** 3 good
**Contribution:** 2 fair
**Rating:** 5
**Confidence:** 4

**Summary:**

The author introduces TrojFSL, a prompt-tuning technique for conducting backdoor attacks, comprising three modules: balanced poison learning, selective token poisoning, and Trojan-triggered attention. Experiments across various downstream tasks demonstrate that TrojFSL significantly outperforms previous works in terms of attack success rate.

**Strengths:**

- The motivation behind this study is clear, and the writing is articulate.
- The research on few-shot backdoor attacks in the context of large language models holds significant real-world relevance.
- The paper identifies limitations in prior works, providing valuable insights for subsequent attack designs.

**Weaknesses:**

- The limitations in technical innovation. While the paper outlines existing issues faced by few-shot attacks, such as an imbalanced poisoned dataset, overfitting, and lack of attention awareness, the proposed methods appear to be a combination of various tricks without offering new technical insights. While the effectiveness of the technique is acknowledged, it lacks novelty in terms of technical approaches.
- Excessive hyperparameters requiring adjustment. The three enhancement strategies introduced in the paper require varying degrees of hyperparameters. For instance, adjusting $\beta$ and $\lambda$ in the balanced dataset, controlling token mask $\gamma$ to mitigate overfitting, and managing attention loss updates for attention awareness. The introduction of numerous parameters complicates the tuning process, making practical application challenging.
- The effectiveness of the proposed method in a black-box setting is not addressed. Current large language models are typically accessed through APIs, raising questions about whether the proposed technique can achieve efficient few-shot attacks in a black-box scenario. If not, it is essential to provide necessary discussion and explanations.
- Lack of comprehensive ablation experiment results. Table 6 only presents linear combination results of different strategies. More diverse combinations should be provided to help readers understand the specific effects of each strategy or their combinations. Since the proposed method involves multiple hyperparameters, it is crucial to conduct ablation experiments on more representative datasets to explore the impact of these parameters.

**Questions:**

See weaknesses above.

---

> ### Author Response · Authors · 2023-11-23
> **Reply to Reviewer oqR6**
>
> We thank reviewer oqR6 for their careful and valuable review of our manuscript and for providing constructive feedback.
>
> **Q1: Technical innovation.**
>
> As Table 1 shows, directly applying prior work to a few-shot prompt attack suffers from either low ASR or low accuracy. Identifying new techniques that can simultaneously achieve high ASR and maintain high accuracy presents a significant challenge. To design techniques, it is very important to understand the reasons why a low ASR and low accuracy happen. Thus, our technique novelties not only include the formulation of identified issues: 1. imbalanced poisoned dataset, 2. overfitting, and 3. lack of attention awareness, but also include their corresponding solutions: 1. Balancing the Poisoned Dataset, 2. Selective Token Poisoning, and 3. Trojan-Trigger Attention. These three technique solutions have been developed based on our insights into the inherent characteristics of the issues they address.  For example, before an attack, the target class and a non-target class possess an equal number of samples. However, once poisoned samples are added to the target class, it results in a larger sample count for the target class. This imbalance is particularly pronounced in the case of few-shot samples. And it is not straightforward to balance the poisoned samples for few-shot prompt attacks. Tuning the poisoning ratio cannot solve the problem. We formulate the method into a simple equation 2. We convert a complex problem into a simple format. Similarly, our selective token poisoning is designed to only insert Trojans into the tokens in the soft prompt with the lowest importance score, while the other tokens in the prompt remain untainted. This is a backdoor-prompt co-design to solve the attack overfitting issue. The trojan-trigger attention is designed to increase the attention on triggers instead of clean tokens and prompts, which is novel and significant for enhancing clean accuracy and ASR.
>
> **Q2: Excessive hyperparameters requiring adjustment. For instance, adjusting beta and alpha in the balanced dataset.**
>
> Most parameters already exist in prior backdoor works such as the poison ratio $\alpha$ and the coefficient $\lambda$. The parameters we introduced in the paper are dataset balance coefficient $\beta$, pruned token number $\gamma$, and attention-loss coefficient $\lambda_1$. Notice that $\beta$ is fixed once poison ratio $\alpha$ and the weight coefficient $\lambda$ are known since our balanced dataset in Equation 2 requires $\beta +\lambda \alpha(n-1)=1$, where $n$ is the class number. In our experiments of Table 2,3,6,7, $\beta=0.5$, $\gamma=1$ and $\lambda_1=1$ (We have added it in section 4). One can readily apply this set of parameters in experimental setups. It is important to note that we have conducted detailed studies on these three parameters: Tables 4 and 5 analyze the parameter $\beta$, Table 8 is dedicated to the pruned token number $\gamma$, and Table 9 explores the attention-loss coefficient $\lambda_1$.
>
> **Q3: Black-box setting for APIs? If not, please discuss.**
>
> White-box model attacks are still important. Numerous model-sharing platforms, such as Hugging Face and Model Zoo, facilitate model submissions and downloads for users. Consequently, the exploration of white-box attacks remains crucial. Notably, our work on few-shot prompt attacks addresses a previously unexplored area and white-box few-shot prompting attacks still suffer from several challenges. We identified several unveiling challenges and proposed solutions to solve them, which marked a significant stride in the direction of few-shot prompt attacks.
>
> **Q4: More diverse combinations of techniques and Hyperparameters on more representative datasets.**
>
> We examined various combinations of the proposed techniques in Table A to further show the individual impact of each technique. It is evident that with the inclusion of any proposed technique, both CDA and ASR show an increase.
>
> ### Table A: Various Combinations of TrojFSL Techniques.
> | Scheme                   | ACC (%) | CDA(%) | ASR(%) |
> |--------------------------|--------:|-------:|-------:|
> | Our Baseline Attack      |   78.14 |  56.52 |  94.08 |
> | Selective Token Poisoning|   78.14 |  63.98 |  97.43 |
> | + Trojan-Trigger Attention | 78.14 |  73.64 |  99.04 |
> | + Balanced Poison Learning | 78.14 |  77.47 |  99.27 |
> |||||
> | Our Baseline Attack      |   78.14 |  56.52 |  94.08 |
> | Trojan-Trigger Attention |   78.14 |  69.97 |  99.21 |
> | + Balanced Poison Learning | 78.14 |  72.65 |  99.25 |
> | + Selective Token Poisoning | 78.14 |  77.47 |  99.27 |
>
> Table B shows that  \( \beta = 0.5 \) and \( \alpha = 1 \) works well on the SST-5 dataset too.
>
> ### Table B
>
> | α    | λ    | ACC (%) | CDA (%) | ASR (%) |
> |------|------|--------:|--------:|--------:|
> | 1/8  | 1    |   33.14 |   31.85 |   25.63 |
> | 1/4  | 1/2  |   33.14 |   32.93 |   72.47 |
> | 1/2  | 1/4  |   33.14 |   31.63 |   91.77 |
> | 1    | 1/8  |   33.14 |   32.6  |   98.31 |

---

> > ### Comment · Reviewer_oqR6 · 2023-11-23
> > **Follow-up.**
> >
> > Thank you for addressing some of my concerns. However, I am still worried about effectively setting hyperparameters for the proposed method and its generalization across different language models and datasets. Also, the design of attacks in black-box scenarios is of practical significance. If the author can improve the work in this scenario, I believe it will significantly enhance the impact of the current research. Considering these points, I will maintain the original score.